# The association of cycling with all-cause, cardiovascular and cancer mortality: findings from the population-based EPIC-Norfolk cohort

Shannon Sahlqvist,[1,2,3] Anna Goodman,[2,4] Rebecca K Simmons,[1] Kay-Tee Khaw,[5] Nick Cavill,[6,7] Charlie Foster,[7] Robert Luben,[8] Nicholas J Wareham,[1,2] David Ogilvie[1,2]

▸ Prepublication history and additional material for this paper is available online. To view these files please visit the journal online (http://dx.doi.org/10.1136/bmjopen-2013-003797).

For numbered affiliations see end of article.

Correspondence to
Dr Shannon Sahlqvist;
shannon.sahlqvist@deakin.edu.au

## ABSTRACT

**Objectives:** To investigate associations between modest levels of total and domain-specific (commuting, other utility, recreational) cycling and mortality from all causes, cardiovascular disease and cancer.

**Design:** Population-based cohort study (European Prospective Investigation into Cancer and Nutrition study-Norfolk).

**Setting:** Participants were recruited from general practices in the east of England and attended health examinations between 1993 and 1997 and again between 1998 and 2000. At the first health assessment, participants reported their average weekly duration of cycling for all purposes using a simple measure of physical activity. At the second health assessment, participants reported a more detailed breakdown of their weekly cycling behaviour using the EPAQ2 physical activity questionnaire.

**Participants:** Adults aged 40–79 years at the first health assessment.

**Primary outcome measure:** All participants were followed for mortality (all-cause, cardiovascular and cancer) until March 2011.

**Results:** There were 22 450 participants with complete data at the first health assessment, of whom 4398 died during follow-up; and 13 346 participants with complete data at the second health assessment, of whom 1670 died during follow-up. Preliminary analyses using exposure data from the first health assessment showed that cycling for at least 60 min/week in total was associated with a 9% reduced risk of all-cause mortality (adjusted HR 0.91, 95% CI 0.84 to 0.99). Using the more precise measures of cycling available from the second health assessment, all types of cycling were associated with greater total moderate-to-vigorous physical activity; however, there was little evidence of an association between overall or domain-specific cycling and mortality.

**Conclusions:** Cycling, in particular for utility purposes, was associated with greater moderate-to-vigorous and total physical activity. While this study provides tentative evidence that modest levels of cycling may reduce the risk of mortality, further research is required to confirm how much cycling is sufficient to induce health benefits.

### Strengths and limitations of this study

- The strengths of this study include its prospective design, the inclusion of a large heterogeneous population of men and women and the long follow-up.
- Further, this study used detailed measures of cycling and overall physical activity to examine associations between the various domains of cycling and mortality.
- Owing to the low average levels of cycling, we were not able to examine the specific effects of a higher 'dose' of cycling, and the analyses were underpowered to examine sex differences in the associations between cycling and mortality.

## INTRODUCTION

Promoting cycling as an alternative to motorised transport would result in reduced carbon emissions, traffic congestion and noise pollution while providing people with an opportunity to integrate regular physical activity into their lives.[1 2] As such, there is increasing policy interest in quantifying the health benefits of cycling so that they can be accurately modelled in the economic appraisal of proposed policies and interventions in the transport and health sectors.[3 4] One such tool developed by the WHO (Health Economic Assessment Tool; HEAT) estimates the economic value of a reduction in mortality as a consequence of population increases in cycling.[5] It does so by assuming a linear dose–response relationship between cycling and mortality and that any increase in cycling is in addition to other physical activity.

HEAT model estimates are dependent on the use of a relative risk estimate from a single study of Danish adults. The study reported a 28% reduction in all-cause mortality in adults who cycled to and from work

compared with those who did not, even after controlling for other physical activity.[6] Similarly, an inverse association between transport (utility) cycling more generally and all-cause and cancer mortality has been reported in a cohort of Chinese women.[7] These findings are likely to reflect, in part, the fact that utility cycling translates into greater overall physical activity.[8 9]

While these studies suggest substantial health benefit associated with utility cycling, an examination of the benefits of recreational cycling would also be valuable to enable more informed policy recommendations on which type of cycling to promote.

Furthermore, it is possible that the findings from these studies reflect, at least to some extent, residual confounding from 'other' physical activity. In particular, the Danish study controlled for recreational physical activity using responses to a single item which asked participants to select from one of four options ranging from 'you are almost entirely sedentary or perform light physical activity less than 2 h/week' to 'you perform highly vigorous physical activity more than 4 h/week or regular exercise or competitive sports several times per week'.[6] The extent to which responses to this item were independent of those regarding commuter cycling was not reported.

In addition, in the two prior studies which reported associations between utility cycling and mortality, the time spent cycling for transport in the exposed groups was substantial, reflecting the relatively high levels of cycling in those countries. For example, in the Danish study, those who commuted by bike spent an average of 180 min/week doing so.[6] In the study of Chinese women, 19% cycled for up to 3.4 metabolic equivalents (MET) h/day while a further 5% cycled for greater than 3.5 MET h/day, equivalent to approximately 350 min/week.[7] Few studies have examined associations between cycling and mortality in populations such as that of the UK, which have a low prevalence of utility cycling by international standards. One previous study of adults in the European Prospective Investigation into Cancer and Nutrition study (EPIC)-Norfolk cohort found no significant association between commuter cycling and either cardiovascular or all-cause mortality.[10] These null findings may partly reflect the cut points used to define cycling categories: the cut point for the highest category was 30 min/week, which may be an insufficient 'dose' to induce health benefits. It is also possible that the relatively short duration of follow-up (7 years) and the small number of deaths in the cohort limited the power of the study to detect effects.

Building on these previous analyses of EPIC-Norfolk cohort data, this paper aims to investigate more comprehensively the mortality benefits of cycling. First, we use a simple pragmatic measure of physical activity to examine associations between total cycling and all-cause, cardiovascular and cancer mortality over 15 years. Second, using a more detailed, disaggregate measure of physical activity which provides more accurate estimates of domain-specific cycling (commuting, all utility and recreational) for a subset of our sample, we explore whether this association is driven by particular domains of cycling (eg, utility vs recreational). Finally, to help explain any associations between domain-specific cycling and mortality, we examine associations between these domains of cycling and total physical activity.

## METHODS
### Study design and participants
This study uses data from the EPIC-Norfolk cohort, part of the 10-country collaborative EPIC. Between 1993 and 1997, 25 633 adults aged 40–79 years were recruited from general practices in the county of Norfolk in the east of England and attended a health examination. As part of this examination, participants completed a short physical activity questionnaire which asked about time spent walking and cycling for all purposes and time spent in other exercise.[11] Between January 1998 and October 2000, 15 519 (61%) of the original cohort attended a second health assessment, completing a more detailed questionnaire on recreational, occupational, utility and household physical activity (EPAQ2).[12]

Data from the first health assessment were used to examine the association between total cycling and cardiovascular disease, while data from the second health assessment were used to examine the association between the domains of cycling and cardiovascular disease. Full details of the study are reported elsewhere.[13]

Of the participants in the first health assessment, we excluded those with self-reported cardiovascular disease (n=1102) or cancer (n=1327) and those with missing data (n=784) leaving 22 450 for analysis.

Similarly, of those who returned for the second health assessment, we excluded those with self-reported cardiovascular disease (n=772) or cancer (n=1115) and those with missing data (n=286), leaving 13 346 for analysis. All participants were followed up for mortality to 31 March 2011 (mean 15.3 years (SD=3.3) from first health assessment, mean 11.5 years (SD=2.0) from second health assessment).

### Health assessments
At both health assessments, participants reported their level of education (categorised as no formal qualification; GCSE or equivalent, that is, examinations normally taken at age 16; 'A' level or equivalent, that is, examinations normally taken at age 18; university degree or equivalent), paid employment status (yes, no), social class (categorised as professional, managerial/technical job, skilled/partially skilled labour, unskilled labour), smoking status (current, former, never), antihypertensive medication (yes, no), medication for dyslipidaemia (yes, no) and family history of cancer and cardiovascular disease (yes, no). History of myocardial infarction, stroke and cancer were also reported. Total energy intake (kJ/day) and alcohol consumption (units/week)

were derived from a validated 130-item semiquantitative food-frequency questionnaire.[14]

### Measurement of physical activity at first health assessment

Physical activity was assessed by asking participants to report, separately for winter and summer, the weekly time (in hours) spent walking and cycling (separately) to work and during leisure, and in other exercise.[11] Total cycling was calculated as the average weekly time spent in winter and summer (min/week; (see online supplementary appendix, part 1).

### Measurement of physical activity at second health assessment

Physical activity, including cycling, was assessed with the validated and reliable EPAQ2 questionnaire, which asks participants to recall their physical activity behaviour across the domains of household, work, recreation and commuting, over the past year.[12] Energy expenditure [MET h/week] was calculated using the physical activity compendium.[15] Following standard EPAQ2 data

reduction rules, we calculated four specific cycling measures explained in detail in table 1. In addition, total moderate-to-vigorous physical activity was calculated as the sum of all moderate and vigorous physical activity across all domains (home, work, recreation and commuting; MET h/week) and recreational physical activity was computed as the sum of all moderate and vigorous activity done during recreation specifically (MET h/week). A copy of the questionnaire can be found at: http://www.srl.cam.ac.uk/epic/questionnaires/epaq2/epaq2.pdf.

### Cycling exposure

Given the highly skewed nature of the cycling data and to allow for comparisons with previous studies, we created three categories of cycling exposure: 0, 1–59 and ≥60 min/week. These represent levels of cycling which we believe are realistic to achieve in countries such as the UK, which currently have low levels of utility cycling. For our measures of utility cycling from the second health assessment, these categories are equivalent to: 0; 0.01–9.99 and ≥10 miles/week.

**Table 1** Cycling exposure measures calculated from the EPAQ2 questionnaire administered at the second health assessment

| Exposure | Calculation |
| --- | --- |
| Commuter cycling | Respondents were asked how frequently they normally travelled to work by car, public transport, bike or on foot (response options were 'always', 'usually', 'occasionally' or 'never/rarely'). Responses were converted to fractions (always=1, usually=0.75, occasionally=0.25, never/rarely=0). Participants reported the distance between home and work and the average number of times per week they made this journey (multiplied by two to account for the return journey). When cycling was the only mode selected, the total weekly distance cycled was calculated by multiplying the distance from home to work by the number of journeys made. When cycling was selected alongside other modes, the distance cycled was weighted according to the frequency of cycling relative to the frequency of the other modes reported. For example, if a respondent selected 'always' for cycling and driving, it was assumed that cycling accounted for 10%, and driving for 90% of the distance travelled. The total number of journeys was then multiplied by the weighted distance travelled (miles/week) |
| Non-commuting utility cycling | Respondents were asked to recall the average number of journeys they made by bicycle to get about apart from going to work for each of the following distances: 'less than 0.5 miles', '0.5 mile–1.5 miles', '1.5–2.5 miles', '2.5–3.5 miles', '3.5–5.5 miles', and 'more than 5.5 miles'. The total weekly distance travelled was computed by multiplying the reported number of trips by the midpoint value of each distance category (assumed to be 0.25 for <0.5 mile and 6 for >5.5 miles). These values were then summed to provide a measure of distance travelled (miles/week) |
| All utility cycling | Distance travelled for non-commuting utility cycling was added to the distance travelled for commuting cycling to derive a measure of total utility cycling (miles/week) |
| Recreational cycling | Respondents reported the average time spent 'cycling for pleasure' per session and the frequency of such sessions: 'none', 'less than once a month', 'once a month', '2–3 times a month', 'once a week', '2–3 times a week', '4–5 times a week', or 'everyday'. The average weekly cycling duration was computed by converting the frequency into a weekly numerical value (eg, 0.5/52 for 'less than once a month' and (2.5×12)/52 for '2–3 times per month'). The time spent cycling (min/week) was computed by multiplying the average session duration by the average weekly frequency |
| Total cycling | To enable a measure of total cycling to be derived and to allow for comparisons with previous studies, the distance travelled for utility cycling was converted into an estimated duration. Based on self-report data from a recent study of UK adults, we assumed an average cycling speed of 10 miles/h.[27] A measure of total time spent cycling (min/week) was derived by summing time spent in commuting, other utility and recreational cycling |

## Mortality outcomes

All EPIC-Norfolk participants were flagged for death certification with the UK Office of National Statistics (ONS). Trained nosologists coded death certificates according to the ICD-9 or ICD-10 codes. Cardiovascular death was defined as ICD 410–448 (ICD 9) or ICD I10–I79 (ICD 10) as underlying cause of death, which comprises coronary heart disease (410–414 (ICD 9) or I20–I25 (ICD 10)), stroke (430–438 (ICD 9) or I60–I69 (ICD 10)), cardiac failure (428 (ICD 9) or I50 (ICD 10)) and other vascular causes. Cancer death was defined as ICD 140–208 (ICD9) or ICD C00–C97 (ICD10) as the underlying cause.

## Statistical analysis

We used exposure data from the first health assessment to examine associations between total cycling and all-cause, cardiovascular and cancer mortality. Exposure data from the second health assessment were used to explore associations between total and domain-specific cycling and mortality and overall physical activity. First, using data collected from the first health assessment, we examined preliminary associations between total cycling and all-cause, cardiovascular and cancer morality by fitting Cox proportional hazard regression models to estimate HRs and 95% CIs. We first adjusted for sex, age, education level and social class (model A) and then further adjusted for smoking status, family history of cardiovascular disease or cancer, as well as time spent walking and in other exercise. As sensitivity analyses, we ran a further two models. In the first model, we adjusted for weekly alcohol consumption and calorie intake; 4% (n=912) of participants had missing data for these variables. In the second, we further adjusted for medication (hypertension and dyslipidaemia) and type 2 diabetes as we thought it possible that they could be mediating variables on the causal pathway between cycling and mortality. The results of these subgroup analyses did not differ substantially from those of model B and are not presented. Models were also run after excluding participants who died within 2 years of follow-up (n=181) to minimise the potential effect of reverse causality. This made no substantive differences to the findings (data not presented).

We then examined the associations between the domains of cycling and mortality, again by fitting Cox proportional hazard regression models. Equivalent models to those described above were run except that model B also controlled for all other physical activity energy expenditure (calculated as the sum of all energy expenditure in all domains of physical activity minus that of the respective cycling behaviour). To account for the potentially conservative estimates of commuting cycling undertaken when cycling was selected alongside other modes (see table 1), by way of sensitivity analysis we applied an alternative assumption that commuter cycling was done for 30% (rather than 10%) of these journeys. The findings remained largely unchanged when using these new estimates. Again, our results were substantively unchanged after adjusting for weekly alcohol consumption and calorie intake, or after excluding the 102 participants who died within 2 years of follow-up (data not presented).

For all models, the proportional hazard assumption was verified using the Schoenfeld residuals and Kaplan-Meier plots for all three outcomes. For all models, we also present p values for linear trend, calculated by entering the domains of cycling as continuous rather than categorical variables.

To examine whether any observed associations between cycling and mortality could be explained by differences in overall levels of physical activity, we examined associations between the domains of cycling and physical activity (total leisure-time, total moderate-to-vigorous across all domains, and total light, moderate and vigorous across all domains) by fitting linear regression models with time spent cycling (total and subdomains) as the exposure variables and time spent in (A) recreational and (B) total moderate-to-vigorous physical activity (MET h/week) as the outcome variables controlling for sex, age, social class and highest level of education. All analyses were conducted using STATA, V.12.0 (Stata Corp, Texas, USA).

## RESULTS

### Participant characteristics

At the first health assessment, participants had a mean age of 58 years (SD=19) and just over half were women (55%). Twenty four per cent of the participants reported cycling for a mean of 165 min/week (SD=246). Sociodemographic characteristics of the cohort by cycling status (yes, no) are described in table 2 (for further details of the baseline characteristics of the sample, see online supplementary appendix, part 2, table A1). Respondents who reported any cycling were, on average, younger and more likely to be men. Respondents with no formal qualification were also more likely to cycle compared with respondents with GCSE-level qualifications, while those in skilled or unskilled labour were more likely to cycle than professionals.

By the second health assessment, participants had a mean age of 62 (SD 9) years; just over half were women (57%). Thirty per cent (n=4030) reported any cycling. Of those who cycled, 62% (n=2808) reported cycling for recreation and 72% (n=3269) reported cycling for utility purposes with 26% (n=862) of these reporting commuting cycling. The average cyclist spent 83 min/week cycling. Those who commuted by bicycle spent an average of 61 min/week doing so, while those who cycled for recreation spent an average of 58 min/week doing so. Again, men and those who were younger were more likely to cycle. In addition to the sociodemographic associations observed in data from the first health assessment, respondents working in a managerial/technical position were less likely to cycle than

**Table 2** Descriptive characteristics of sample (N (%) at first health assessment (n=22 450) by cycling (yes, no)

| Characteristic | 0 min/week: N (column %) | ≥1 min/week: N (column %) | OR for any cycling (95%CI)* |
|---|---|---|---|
| **Sex** | | | |
| Male | 3880 (66.9) | 1920 (72.0) | 1.0 |
| Female | 5436 (33.1) | 2110 (28.0) | 0.92 (0.86, 0.98) |
| **Age (years)** | | | |
| 40–55 | 2096 (58.1) | 1498 (41.7) | 1.0 |
| 50–65 | 3105 (67.4) | 1473 (32.2) | 0.72 (0.67, 0.78) |
| ≥65 | 4115 (79.4) | 1059 (20.5) | 0.44 (0.40, 0.49) |
| **Education level** | | | |
| Degree or equivalent | 1280 (65.6) | 670 (34.4) | 1.0 |
| 'A' level or equivalent | 3867 (69.2) | 1719 (30.8) | 0.91 (0.81, 1.02) |
| GCSE or equivalent | 1054 (71.0) | 431 (29.0) | 1.02 (0.95, 1.11) |
| No formal qualification | 3115 (72.09) | 1210 (28.0) | 1.29 (1.15, 1.44) |
| **Social class** | | | |
| Professional | 665 (66.4) | 336 (33.6) | 1.0 |
| Managerial/technical | 5269 (71.8) | 2071 (28.2) | 0.90 (0.79, 1.02) |
| Skilled/partially skilled labour | 3078 (67.7) | 1469 (32.3) | 1.15 (1.00, 1.31) |
| Unskilled labour | 304 (66.4) | 154 (33.6) | 1.36 (1.08, 1.64) |
| **Paid employment** | | | |
| No | 8578 (81.0) | 2080 (19.0) | 1.0 |
| Yes | 8365 (72.5) | 2127 (27.2) | 1.05 (0.97, 1.14) |

*Adjusted for all other variables in the table.

professionals. Employment status also showed a strong association with cycling, probably reflecting the fact that commuting was included in the measure of cycling (see online supplementary appendix, part 2, table A2).

### Total cycling (first health assessment) and mortality

In total, 4398 (20%) participants died during 3 425 498 person-years of follow-up (table 3). There were 1379 (6.1%) cardiovascular deaths and 1639 (7.3%) cancer deaths (see online supplementary appendix, Part 2, table A3). Risk of death was associated with being male older and having a lower level of education and social class.

Cycling for at least 60 min/week was associated with a 9% reduction in all-cause mortality after controlling for potential confounders (HR 0.91, 95% CI 0.84 to 0.99; table 3). In the minimally adjusted model, cycling for at least 60 min/week was associated with a 19% reduction in cardiovascular mortality (HR 0.81, 95% CI 0.69 to 0.95); however, this was no longer significant after controlling for potential confounders including time spent walking and in other exercise. Cycling was not associated with cancer mortality.

### Domains of cycling (second health assessment) and mortality

In total, 1670 (12.5%) individuals died during 149 072 person-years of follow-up. There were 485 (3.6%) cardiovascular deaths and 700 (5.2%) cancer deaths. Again, mortality rates were higher among men and older participants (data not shown). There were no significant associations between commuting cycling and all-cause, cardiovascular or cancer mortality in either the minimally adjusted (A) or the additionally adjusted (B) models (table 4). For all-cause and cancer mortality, however, there was a suggestion of a dose–response relationship between the distance cycled and risk of death whereby the lowest HRs were observed for the highest levels of commuting cycling, albeit not reaching statistical significance. There was no association between all utility cycling and all-cause, cardiovascular or cancer mortality in either the minimally adjusted (A) or the additionally adjusted (B) models. In minimally adjusted models, recreational cycling for less than 60 min/week was associated with a 19% (95% CI 0.66 to 0.99) reduction in risk. Further adjustment attenuated the effect. There were no significant associations between total cycling and mortality.

### Association between domains of cycling and total physical activity (second health assessment)

Total and domain-specific cycling was associated with greater levels of physical activity in an approximately dose–response relationship (table 5). All utility, recreation and total, but not commuting, cycling was associated with greater recreational physical activity. Importantly, however, commuting cycling was not inversely associated with recreational physical activity, suggesting that adults were not cycling to and from work to compensate for a lack of recreational physical activity. The association between cycling and recreational physical activity was strongest for recreational cycling; those who spent 1–59 min/week cycling for pleasure participated in an additional 3 MET h/day of recreational physical activity (equivalent to approximately 36 min/day of moderate intensity physical activity).

**Table 3** Prospective associations over 15 years between total cycling and mortality (all-cause, cardiovascular and cancer) in 22 450 participants

| | | All-cause mortality | | | Cardiovascular mortality | | | Cancer mortality | | |
|---|---|---|---|---|---|---|---|---|---|---|
| | | | HR (95%CI) | | | HR (95%CI) | | | HR (95%CI) | |
| Total cycling | FU years Mean (SD) | Events N (%) | Model A† | Model B‡ | Events N (%) | Model A† | Model B‡ | Events N (%) | Model A† | Model B‡ |
| 0 min/week | 15.2 (3.4) | 3686 (21.3) | 1 | 1 | 1179 (6.8) | 1 | 1 | 1352 (7.8) | 1 | 1 |
| 1–59 min/week | 15.7 (2.6) | 100 (10.8) | 0.86 (0.71, 1.07) | 0.96 (0.78, 1.17) | 25 (2.7) | 0.73 (0.49, 1.08) | 0.83 (0.56, 1.24) | 44 (4.7) | 0.91 (0.68, 1.24) | 0.99 (0.73, 1.34) |
| ≥60 min/week | 15.7 (3.0) | 612 (14.3) | 0.86 (0.79, 0.94)** | 0.91 (0.84, 0.99)* | 179 (4.2) | 0.81 (0.69, 0.95)* | 0.87 (0.74, 1.02) | 252 (5.9) | 0.89 (0.77, 1.01) | 0.93 (0.81, 1.06) |
| p for linear trend | | | 0.02 | 0.06 | | 0.04 | 0.09 | | 0.20 | 0.23 |

*p<0.05, **p<0.01, ***p<0.001.
†Adjusted for age, sex, education level and social class.
‡Further adjusted for smoking status, family history of cancer or cardiovascular disease, and other physical activity (walking and other exercise).
FU, follow-up.

All domains of cycling showed significant dose-response relationships with total moderate-to-vigorous physical activity, although the association was strongest for commuting cycling. Those who cycled for ≥60 min/week spent an additional 7.9 MET h/day in moderate-to-vigorous physical activity (equivalent to 94.8 min/day of moderate intensity physical activity) compared with those who did not.

## DISCUSSION

We used data from a large population-based cohort to examine the associations between total and domain-specific cycling and mortality. Across all domains, cyclists were more likely to be younger and men, a finding that is consistent with previous studies conducted in countries that have low rates of utility cycling[16–18] but different from the pattern in a number of other European countries where men and women, and the young and old, are equally likely to cycle.[19] An important finding was that cycling, in particular commuting cycling, was associated with participation in greater levels of total physical activity. These findings support an increasing body of work which shows that active travel is done in addition to, rather than instead of, recreational physical activity.[8 9 20 21] Given the time people spend travelling, and the fact that a shift from motorised to active travel may result in environmental and economic benefit, encouraging participation in cycling appears a valuable way to increase participation in overall physical activity.

Using exposure data from the first health assessment, cycling for at least 60 min/week in total was associated with a 9% reduction in risk of all-cause mortality but was not associated with reductions in risk of cardiovascular or cancer mortality. In the absence of any directly comparable data on total cycling from other studies, these findings provide tentative evidence that modest 'doses' of cycling may be associated with a reduction in mortality risk. They are also broadly consistent with the findings of the Danish study in which a reduction in mortality risk (28%) was associated with an average quantity of cycling that was three times higher (180 min/week).[6]

That being said, when using more precise measures of cycling, we found no significant associations between total or domain-specific cycling and mortality. On the one hand, these differences may reflect the more precise measures of physical activity used in the second health assessment, which may have not only enabled more accurate categorisation of cycling exposure, but also reduced measurement error regarding the confounding effect of 'other' physical activity. On the other hand, they could reflect a lack of power in analyses of the second health assessment data, which included fewer participants and had a shorter follow-up period.

Despite 5 additional years of follow-up and the examination of a higher 'dose' of cycling, our null findings

**Table 4** Prospective association over 11.5 years between cycling (total and domain specific) and mortality (all-cause, cardiovascular and cancer) in 13 346 participants

| | | All-cause mortality | | | Cardiovascular mortality | | | Cancer mortality | | |
|---|---|---|---|---|---|---|---|---|---|---|
| | FU years Mean (SD) | Events N (%) | HR (95%CI) | | Events N (%) | HR (95%CI) | | Events N (%) | HR (95%CI) | |
| | | | Model A† | Model B‡ | | Model A† | Model B‡ | | Model A† | Model B‡ |
| **Commuting** | | | | | | | | | | |
| 0 miles/week (0 min/week) | 11.1 (2.0) | 1630 (13.1) | 1 | 1 | 474 (3.8) | 1 | 1 | 679 (5.4) | 1 | 1 |
| 0.01–9.99 miles/week (1–59 min/week) | 11.5 (1.6) | 29 (4.9) | 0.87 (0.60, 1.26) | 0.96 (0.66, 1.39) | 9 (1.5) | 1.09 (0.56, 2.13) | 1.23 (0.63, 2.40) | 16 (2.7) | 0.90 (0.55, 1.49) | 0.95 (0.57, 1.57) |
| ≥10 miles/week (≥60 min/week) | 11.5 (1.4) | 11 (4.0) | 0.80 (0.44, 1.46) | 0.91 (0.50, 1.65) | 2 (0.7) | 0.61 (0.15, 2.47) | 0.71 (0.18, 2.90) | 5 (1.8) | 0.66 (0.27, 1.59) | 0.68 (0.28, 1.66) |
| p for linear trend | | | 0.42 | 0.74 | | 1.00 | 0.72 | | 0.28 | 0.34 |
| **All utility** | | | | | | | | | | |
| 0 miles/week 0 min/week) | 11.1 (2.0) | 1383 (13.2) | 1 | 1 | 392 (3.8) | 1 | 1 | 580 (5.5) | 1 | 1 |
| 0.01–9.99 miles/week (1–59 min/week) | 11.6 (1.8) | 233 (10.4) | 0.90 (0.78, 1.04) | 0.95 (0.83, 1.09) | 75 (3.4) | 1.04 (0.81, 1.34) | 1.10 (0.85, 1.41) | 97 (4.3) | 0.85 (0.69, 1.06) | 0.90 (0.72, 1.11) |
| ≥10 miles/week (≥60 min/week) | 11.4 (1.8) | 54 (8.4) | 1.01 (0.77, 1.33) | 1.10 (0.83, 1.44) | 18 (2.8) | 1.30 (0.81, 2.10) | 1.44 (0.89, 2.31) | 23 (3.6) | 0.89 (0.58, 1.35) | 0.92 (0.61, 1.41) |
| p for linear trend | | | 0.71 | 0.33 | | 0.81 | 0.52 | | 0.94 | 0.89 |
| **Recreational** | | | | | | | | | | |
| 0 min/week | 11.1 (2.1) | 1483 (13.7) | 1 | 1 | 438 (4.0) | 1 | 1 | 608 (5.6) | 1 | 1 |
| 1–59 min/week | 11.4 (1.5) | 104 (5.9) | 0.81 (0.66, 0.99)* | 0.87 (0.69, 1.04) | 25 (1.4) | 0.72 (0.48, 1.09) | 0.75 (0.50, 1.13) | 56 (3.2) | 0.90 (0.68, 1.19) | 0.95 (0.71, 1.25) |
| ≥60 min/week | 11.3 (1.8) | 83 (11.1) | 1.13 (0.90, 1.41) | 1.25 (0.99, 1.55) | 22 (3.0) | 1.07 (0.69, 1.65) | 1.19 (0.77, 1.84) | 36 (4.8) | 1.06 (0.75, 1.49) | 1.12 (0.80, 1.58) |
| p for linear trend | | | 0.12 | 0.05 | | 0.48 | 0.32 | | 0.46 | 0.35 |
| **Total cycling** | | | | | | | | | | |
| 0 min/week | 11.1 (2.1) | 1308 (14.0) | 1 | 1 | 379 (4.1) | 1 | 1 | 540 (5.8) | 1 | 1 |
| 1–59 min/week | 11.5 (1.8) | 236 (8.9) | 0.87 (0.76, 1.00)* | 0.90 (0.79, 1.04) | 72 (2.7) | 0.95 (0.74, 1.23) | 0.99 (0.76, 1.23) | 105 (4.0) | 0.86 (0.70, 1.06) | 0.90 (0.73, 1.11) |
| ≥60 min/week | 11.4 (1.7) | 126 (9.2) | 1.00 (0.83, 1.21) | 1.11 (0.91, 1.32) | 34 (2.5) | 1.00 (0.70, 1.43) | 1.10 (0.77, 1.57) | 55 (4.0) | 0.92 (0.69, 1.22) | 0.98 (0.74, 1.30) |
| p for linear trend | | | 0.08 | 0.26 | | 0.36 | 0.18 | | 0.81 | 0.51 |

*p<0.05, **p<0.01, ***p<0.001
†Adjusted for age, sex, education level and social class.
‡Further adjusted for smoking status, family history of cancer or cardiovascular disease, and all other physical activity.
FU, follow-up.

**Table 5** Associations between time spent cycling (total and subdomains; min/week) and physical activity (MET h/week) in 13 346 participants

| | N | Leisure time PA† (MET h/week) | | Total MVPA‡ (MET h/week) | | Total PA§ (MET h/week) | |
|---|---|---|---|---|---|---|---|
| | | Mean (SD) | Regression coefficient (95% CI)¶ | Mean (SD) | Regression coefficient (95% CI)¶ | Mean (SD) | Regression coefficient (95% CI)¶ |
| Commuter cycling | | | | | | | |
| 0 miles/week (0 min/week) | 12 484 | 39.4 (37.7) | 0 | 61.9 (52.3) | 0 | 82.1 (44.6) | 0 |
| 0.01–9.99 miles/week (1–59 min/week) | 587 | 35.3 (32.4) | −1.6 (−4.7, 1.4) | 82.4 (58.0) | 12.2 (8.2, 16.2)*** | 104.9 (46.8) | 11.8 (8.3, 15.3)*** |
| ≥10 miles/week (≥60 min/week) | 275 | 42.9 (37.6) | 1.9 (−2.5, 6.3) | 116.6 (63.0) | 35.3 (29.5, 41.0)*** | 128.6 (49.2) | 29.7 (24.7, 34.7)*** |
| All utility cycling | | | | | | | |
| 0 miles/week (0 min/week) | 10 462 | 38.2 (37.1) | 0 | 60.5 (51.7) | 0 | 81.2 (44.2) | 0 |
| 0.01–9.99 miles/week (1–59 min/week) | 2237 | 42.9 (38.5) | 3.7 (2.1, 5.4)*** | 69.6 (55.4) | 5.5 (3.3, 7.7)*** | 89.2 (46.9) | 4.4 (2.5, 6.3)*** |
| ≥10 miles/week (≥60 min/week) | 647 | 46.7 (39.3) | 6.9 (3.9, 9.8)*** | 99.0 (60.4) | 25.0 (21.2, 28.9)*** | 112.1 (50.0) | 20.2 (16.9, 23.6)*** |
| Leisure-time cycling | | | | | | | |
| 0 min/week | 10 843 | 37.3 (36.5) | 0 | 60.0 (51.7) | 0 | 80.5 (44.1) | 0 |
| 1–59 min/week | 1756 | 41.0 (35.4) | 4.2 (2.3, 6.1)*** | 73.8 (53.9) | 4.7 (2.3, 7.2)*** | 95.3 (45.3) | 4.9 (2.8, 7.0)*** |
| ≥60 min/week | 747 | 64.6 (46.6) | 25.6 (22.9, 28.2)*** | 98.0 (61.5) | 27.4 (23.9, 31.0)*** | 110.2 (52.5) | 21.9 (18.9, 25.0)*** |
| Total cycling | | | | | | | |
| 0 min/week | 9316 | 37.2 (36.6) | 0 | 58.5 (50.8) | 0 | 79.3 (43.6) | 0 |
| 1–59 min/week | 2654 | 39.7 (35.8) | 3.2 (1.6, 4.9)*** | 67.0 (53.1) | 2.9 (0.8, 5.0)* | 88.3 (45.0) | 2.6 (0.8, 4.4)* |
| ≥60 min/week | 1376 | 52.4 (43.6) | 14.2 (12.2, 16.3)*** | 94.4 (60.5) | 24.1 (21.3, 26.8)*** | 108.7 (50.7) | 19.5 (17.1, 21.8)*** |

*p<.05, **p<0.01, ***p<.001.
†Computed as the sum of all moderate-to-vigorous leisure-time physical activity.
‡Computed as the sum of all moderate-to-vigorous physical activiy across all domains (leisure-time, household, work, commute).
§Computed as the sum of all light and moderate-to-vigorous physical activity across all domains (leisure-time, household, work, commute).
¶Adjusted for age, sex, education and social class.
MVPA, moderate-to-vigorous physical activity; PA, physical activity.

relating to the mortality benefits of commuting and utility cycling in particular mirror those previously reported in this cohort[10] and are consistent with those of previous studies of low-cycling populations in Northern Ireland and France, which found no evidence of a reduced risk of fatal or non-fatal myocardial infarction in men who reported any walking or cycling to work compared with those who did not.[22]

They are, however, in contrast to the findings of the studies of Danish[6] and Chinese[7] adults and of a meta-analysis, which pooled evidence from eight studies (from five independent populations) and found that active commuting (walking and cycling) was associated with an overall 11% reduction in the risk of cardiovascular outcomes.[23] Importantly, the levels of commuting cycling reported by participants in these previous studies were substantial and in the meta-analysis, evidence of protective effects was generally limited to higher levels of active commuting.[23] The high 'doses' of utility cycling reported in previous studies are likely to be achieved

when cycling journeys are taken frequently and consistently (eg, twice daily, 5 days/week). It is possible that frequent short bursts of physical activity of this kind are beneficial to health in their own right, rather than simply by contributing to greater levels of total physical activity as we have shown. In support of this hypothesis, studies have demonstrated that accumulated short bouts of exercise over the day result in longer postexercise reductions in blood pressure[24] and lower plasma triglycerides[25] than one continuous session of exercise. There is also some evidence that the intensity of cycling is important. A study of Danish adults found a significant inverse association between cycling intensity and all-cause and coronary heart disease mortality,[26] and it may be that participants in our study were not cycling at an intensity sufficient to result in health benefit. It is also possible that the differences reflect the fact that our cohort was older than the Danish and Chinese cohorts.

To further elucidate the health benefits of cycling and refine the use of tools such as HEAT that may be used to

inform policy in this area, future research should aim to estimate the association between cycling and mortality independent of other physical activity, measured with as little error as possible; to extend such analyses to include morbidity endpoints such as incident cardiovascular disease and diabetes; and to clarify how much cycling is sufficient to induce health benefits by quantifying the mean quantity (and preferably intensity) of cycling in each exposure category studied and describing the shape of the dose–response relationship. In the meantime, our results suggest that even modest 'doses' of cycling *may* reduce mortality risk and do not suggest any evidence of an adverse effect, thereby contributing to the growing environmental, social and public health case for promoting cycling in individuals and populations.

## STRENGTHS AND LIMITATIONS

This is the first study to examine independent associations between total and domain-specific cycling and mortality. The other strengths of the study include its prospective design, the inclusion of a large heterogeneous population of men and women and its long follow-up. We adjusted our analyses of the second health assessment data for all types of physical activity as well as a range of potential demographic and behavioural confounders, which strengthens the inferences made. Excluding participants with existing chronic disease and those who died within two years of follow-up enabled us to control for reverse causality. Given the population-based recruitment from a large geographical area, we believe that our findings are generalisable to middle-aged and older-aged adults. There are, however, a number of limitations. Cycling and total physical activity were assessed by self-report. The cycling exposure variables, in particular utility cycling, were derived from relatively crude measures and assumptions had to be made about frequency of cycling, distance travelled and average speed. Owing to the low average levels of cycling, we were not able to examine the specific effects of a high 'dose' of cycling and the analyses were underpowered to examine sex differences in the associations which have been previously documented.[23]

## CONCLUSIONS

Building on previous research that demonstrated inverse associations between high doses of utility cycling and mortality, we used data from a large population-based cohort to examine associations between more modest levels of cycling and mortality. Cycling, in particular for utility purposes, was associated with greater levels of total and moderate-to-vigorous physical activity. This was largely due to the fact that adults who cycled did not participate in less leisure-time physical activity. Despite these positive associations, there was little evidence that cycling was associated with a reduction in mortality risk. While our preliminary findings suggest that low levels of cycling are associated with a reduced risk of mortality, these findings were not replicated when using more detailed measures of exposure, albeit in fewer participants who were followed up for a shorter period. Nevertheless, cycling provides an opportunity to incorporate frequent physical activity into activities of daily living, and when done as a means to get from place to place, it may also provide substantial environmental and economic benefits to society.

**Author affiliations**
[1]Medical Research Council Epidemiology Unit, University of Cambridge, Cambridge, UK
[2]UKCRC Centre for Diet and Activity Research (CEDAR), University of Cambridge, Cambridge, UK
[3]Centre for Physical Activity and Nutrition Research (C-PAN), School of Exercise and Nutrition Sciences, Deakin University, Geelong, Australia
[4]Faculty of Epidemiology and Population Health, London School of Hygiene and Tropical Medicine, London, UK
[5]Department of Gerontology in Clinical Medicine, University of Cambridge, Cambridge, UK
[6]Cavill Associates, Stockport, UK
[7]British Heart Foundation Health Promotion Research Group, Nuffield Department of Population Health, University of Oxford, Oxford, UK
[8]Department of Public Health and Primary Care, Institute of Public Health, University of Cambridge School of Clinical Medicine, Cambridge, UK

**Acknowledgements** The authors thank Dr Søren Brage and Dr Marcel Den Hoed for their assistance in cleaning the EPAQ2 data. The authors thank the EPIC-Norfolk staff and participants for their valuable contributions.

**Correction notice** This article has been corrected since it was published Online First. The open access statement has been corrected.

**Contributors** DO and CF conceived the idea of the analysis and SS and DO designed the analysis with advice from the other authors. SS analysed the data and wrote the initial draft of the manuscript with DO, AG and RKS. KTK, NJW and RL contributed to the design of the EPIC-Norfolk study protocol. All authors contributed to the critical review of the design of the analysis and the critical revision of the manuscript and approved the final version.

**Funding** DO is supported by the Medical Research Council (Unit Programme number MC_UP_1001/1) and by the Centre for Diet and Activity Research (CEDAR), a UKCRC Public Health Research Centre of Excellence which also hosted SS and AG. The EPIC-Norfolk study is funded by the Cancer Research Campaign, the Medical Research Council, the Stroke Association, the British Heart Foundation, the Department of Health, the Europe Against Cancer Programme Commission of the European Union and Ministry of Agriculture, Fisheries and Food. Funding from the British Heart Foundation, Economic and Social Research Council, Medical Research Council, the National Institute for Health Research, and the Welcome Trust, under the auspices of the UK Clinical Research Collaboration, is gratefully acknowledged. AG is supported by a National Institute of Health Research (NIHR) postdoctoral fellowship.

**Competing interests** None.

**Patient consent** Obtained.

**Ethics approval** The Norwich District Health Authority Ethics Committee approved the study design.

**Provenance and peer review** Not commissioned; externally peer reviewed.

**Data sharing statement** Statistical codes are available on request from Shannon Sahlqvist at shannon.sahlqvist@deakin.edu.au. Researchers wishing to access EPIC data to replicate or extend these analyses should contact Robert Luben (robert.luben@phpc.cam.ac.uk) in the first instance.

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
