## [Reviewer comments · BMJ Open]

Some articles will have been accepted based in part or entirely on reviews undertaken for other BMJ Group journals. These will be reproduced where possible.

ARTICLE DETAILS

TITLE (PROVISIONAL)	The association of cycling with all-cause, cardiovascular and cancer mortality: findings from the population based EPIC-Norfolk cohort
AUTHORS	Sahlqvist, Shannon; Goodman, Anna; Simmons, Rebecca; Khaw, KayTee; Cavill, Nick; Foster, Charles; Luben, Robert; Wareham, Nicholas; Ogilvie, David

VERSION 1 - REVIEW

REVIEWER	Shephard, Roy University of Toronto
REVIEW RETURNED	28-Aug-2013

THE STUDY	The proportion of cyclists seems high for older people in England.
RESULTS & CONCLUSIONS	I think the suggestion of an effect of cycling is not warranted by the data, and more emphasis should be given to the idea that cycling is serving as an index of overall interest in physical activity.
GENERAL COMMENTS	There have been a number of articles and reviews on active commuting and health in recent years, as the authors acknowledge in their list of references. The question is what new information does this study offer. Its assets are a large sample and a substantial (15-year) follow-up period; there is also an attempt to disentangle cycling from an associated propensity to engage in physical activity. The paper is based on exploitation of data collected for a cancer study, and one limitation is the age of the subjects, which averaged 58 years. Relatively few in this age group are likely to cycle, and any relationship between cycling and cardiovascular health may have been set earlier in life. The reported percentage of cyclists (24% at 58 yr, 30% at 62 yr) seems high for this age group, and one wonders about a biased sample. Although the authors claim tentative evidence of benefit from cycling <60 min/day, their Table 3 shows that after adjustment for other types of activity, the benefit is not statistically significant. The same is true of Table 4. The main finding, which accorded with my initial expectations, is that short periods of cycling reflect an overall interest in physical activity (Table 5), and this is the part of the paper that merits being given greater prominence. References- the author of #2 is Shephard

REVIEWER	Rissel, Chris University of Sydney
REVIEW RETURNED	03-Sep-2013

THE STUDY	No further information needs to be included in the manuscript.
RESULTS & CONCLUSIONS	The results may be inconclusive, as they are inconsistent with earlier work, and have a small sample and shorter follow-up period. There is something of a null result, and not entirely clear if there is simply no effect with the low dose of cycling. The authors may need to be more explicit about the null findings and that there may not be a short term mortality benefit with very low levels of cycling, even though there are other policy reasons for encouraging cycling.

VERSION 1 – AUTHOR RESPONSE

Reviewer 1

1. The proportion of cyclists seems high for older people in England

In the absence of any comparable data we cannot say whether or not the proportion of cyclists in our sample is high for older adults living in England. However, the data are drawn from a large, population-based study with a high response rate (for a long-term cohort study). We believe that the estimate is therefore reflective of older adults in the East of England, a region which has the highest rates of cycling in England (Goodman et al., 2013). Furthermore, the definition of cycling included both recreational and utility cycling e.g. participants could have reported one recreational cycling session in the past month to qualify as a “cyclist”.

2. I think the suggestion of an effect of cycling is not warranted by the data, and more emphasis should be given to the idea that cycling is serving as an index of overall interest in physical activity.

In light of this suggestion we have reduced the emphasis on the association between cycling and mortality, and increased the emphasis on the positive association between cycling and overall physical activity.

We have made changes to the Abstract (see lines 51 - 56), the Article Summary (see lines 72 - 75), the Discussion (see lines 285 - 293) and the Conclusion (see lines 349 - 357).

3. There have been a number of articles and reviews on active commuting and health in recent years, as the authors acknowledge in their list of references. The question is what new information does this study offer? Its assets are a large sample and a substantial (15-year) follow-up period; there is also an attempt to disentangle cycling from an associated propensity to engage in physical activity.

We thank the reviewer for highlighting the strengths of our study, all of which are already mentioned in the Introduction and Discussion. While there is increasing evidence of the health benefits of active commuting (in general), there is little evidence of the health benefits of cycling specifically. Moreover, much of the evidence has focused on intermediate health outcomes including cardiovascular disease risk factors. Very few studies have explored the association between cycling and mortality. As we argue in the paper, previous studies include examination of the association between cycling and mortality in populations where there are very high levels of cycling, and results may also reflect residual confounding from engagement in high levels of total physical activity.

4. The paper is based on exploitation of data collected for a cancer study, and one limitation is the age of the subjects, which averaged 58 years. Relatively few in this age group are likely to cycle, and any relationship between cycling and cardiovascular health may have been set earlier in life. The reported percentage of cyclists (24% at 58 yr, 30% at 62 yr) seems high for this age group, and one wonders about a biased sample.

There are strong associations between physical activity performed in middle and older age and health benefit in later life. For example, findings from the Harvard Alumni Health study found that in men (who at baseline had a mean age of 57.5 years, similar to EPIC-Norfolk participants), both maintaining, and taking up an active lifestyle were associated with a 25% reduction in risk of mortality (Paffenbarger et al., 1993). Similar studies have also shown that the uptake of physical activity later in life can reduce the risk of all-cause and cardiovascular mortality (see for example Wannamethee et al., 1998). As life expectancy (though not necessarily quality of life) continues to increase in many parts of the globe, we would argue that it is very important to examine the health benefits of physical activity in middle and older age. The long-term follow-up of the EPIC-Norfolk cohort, alongside the flagging of participants for mortality, affords us a unique opportunity to examine this important public health question in an aging population.

In terms of the percentage of cyclists in EPIC-Norfolk, please see our response to point 1.

5. Although the authors claim tentative evidence of benefit from cycling <60 min/day, their Table 3 shows that after adjustment for other types of activity, the benefit is not statistically significant. The same is true of Table 4. The main finding, which accorded with my initial expectations, is that short periods of cycling reflect an overall interest in physical activity (Table 5), and this is the part of the paper that merits being given greater prominence.

In response to this and point 2 we now discuss in more detail the finding that cycling might be reflective of greater overall physical activity levels (see response to point 2 above).

6. References- the author of #2 is Shephard

Thank you for highlighting this error. It has been amended.

Reviewer 2

1. The results may be inconclusive, as they are inconsistent with earlier work, and have a small sample and shorter follow-up period. There is something of a null result, and not entirely clear if there is simply no effect with the low dose of cycling.

We agree that there may be no effect on mortality with the low dose of cycling seen in our population. However, the “null” result does not mean that our findings do not contribute to the literature or should be withheld from publication, which has the potential to introduce publication bias. Indeed, ours is the first study to examine the association between cycling and mortality in a population with lower levels of cycling than those observed in previous literature. Furthermore, the hazard ratios are in the expected direction of effect, and we would therefore argue that our results are not completely inconsistent with earlier work. While the population is smaller than that reported by Andersen et al., (2000) and Mathews et al.,(2007) our follow-up time is substantially longer than that seen in Matthews et al., (2007) and comparable with Andersen et al., (2000).

2. The authors may need to be more explicit about the null findings and that there may not be a short term mortality benefit with very low levels of cycling though there are other policy reasons for encouraging cycling.

We disagree that our findings examine the short term effect of cycling on mortality. The follow-up periods in this study are comparable to (and in some instances longer than) those of similar studies and are consistent with the follow-up periods of many studies examining the association between general physical activity levels and mortality. We do agree however that there may not be any benefit associated with low levels of cycling, particularly when controlling for participation in other physical

activity. This is an important finding that has policy implications as it suggests that cycling may need to be done at high levels to confer benefit. We agree that despite the lack of a significant association between cycling and mortality seen in this study, there are important economic and environmental benefits that result from encouraging utility cycling. We already allude to these benefits in the Discussion and Conclusion.

We write that:

‘Given the time people spend traveling, and the fact that a shift from motorised to active travel is associated with environmental and economic benefit, encouraging participation in cycling appears a valuable way to increase participation in overall physical activity.’

‘Nevertheless, cycling provides an opportunity to incorporate frequent physical activity into activities of daily living, and when done as a means to get from place to place may also confer substantial environmental and economic benefits to society.’

See page 8 lines, 288-290 and page 10, lines 355-357.

3. This is important research that examines the impact of cycling on mortality from all causes, and specifically from cardiovascular disease and cancer. It is important as a component of improving the costing of the value of cycling (in mortality terms) particularly in a low cycling environment such as the UK.

Thank you for this comment.

4. The results are somewhat surprising. The initial finding (using a fairly general measure of time cycling) that there was a 9% reduction in all-cause mortality associated with cycling an average 60 minutes per week more than 60 minutes a week is not supported by the later analysis (using more precise measures of cycling), although there is a some suggestion of consistency – but no statistical association. The authors highlight that the second analysis has fewer participants and a shorter follow-up, which could explain some differences in the two analyses. This is important, as the shorter follow-up does not allow for sufficient outcome events and reduces the power of the analysis. Is there any scope to increase the length of time for follow-up? The absence of a reduction in cardiovascular mortality associated with more than 60 mins of cycling a week, nor a linear trend is very surprising, but there are very small numbers of CVD events.

We agree that the number of deaths among the smaller number of participants in the second analysis may reduce the power of the analysis. We comment on this point in the Discussion: “could reflect a lack of power in analyses of the second health assessment data, which included fewer participants and had a shorter follow-up period”. Unfortunately, however, there is currently no scope to extend the follow-up analysis beyond March, 2011 until after the next wave of cohort data collection is completed in 2015.

5. They further state that more accurate assessment of cycling and physical activity may have reduced measurement error, and so may accurately represent the effect on mortality of a low ‘dose’ of cycling. If this is true then the conclusion would have to be that low doses of cycling do not directly impact on mortality, and even the tests of trend in table 4 support this conclusion. Or the results are inconclusive. Clearly the authors would like there to be a positive reduction in mortality, but their analysis does not allow them to say it – and they may need to tone down the wording even further than the results “suggest” a reduction in mortality.

Thank you for this comment. In light of this, and other comments made, we have changed the emphasis of the Discussion (see line 285 - 298) and Conclusion (see line 349 - 357).

6. While non-significant, the greatest reductions in mortality appear to be associated with commuting cycling. For me this highlights the issue of whether it is the intensity of the cycling that has the greatest health benefit at low levels of cycling. Does intensity of effort make the difference on health or is it simply how long you do it for (mins per week)?

Recreational cycling can be either intense (road racing) or slow (bike paths) and would have very different effects (but which cancel each other out in the current analysis). Clearly intensity is important, but this has not really been measured.

We already argue in the manuscript that the intensity of cycling may be important. We write that, 'There is also some evidence that the intensity of cycling is important. A study of Danish adults found a significant inverse association between cycling intensity and all-cause and coronary heart disease mortality, and it may be that participants in our study were not cycling at an intensity sufficient to result in health benefit. It is also possible that the differences reflect the fact that our cohort was older than the Danish and Chinese cohorts.'

As it was not possible to measure cycling intensity in this study, we are unable to address this issue in the current paper. As commuting cycling was also associated with total physical activity, we believe that it may not necessarily be the intensity of cycling which is important but the fact that utility cycling is carried out more frequently and in addition to recreational physical activity. The benefits of frequent participation in cycling clearly warrant further investigation following the findings of this paper.

7. If time spent cycling is the important metric, then have the authors considered other cut-points. While one hour a week or more might be an achievable policy target for cycling, it may simply not be enough for demonstrable short term mortality benefits. What about applying the Danish cut point (180 mins) even if the cell sizes become very small in this English sample? Or even 120 minutes? I realise the test for trend was not significant in most of these analysis, but it may not be a linear association, and benefits may only become evident after a minimum amount of time – what is that point? There may need to be more people cycling, and cycling longer, in order to demonstrate this effect.

This is a good point. As we argue in the paper, given that associations between cycling and mortality have only been reported at high volumes of cycling we thought it important to examine associations between lower, more achievable doses of cycling. We did examine the association between 90 and 120 min/week of cycling and mortality. As the reviewer suggests, however, few participants cycled at this volume and the cell sizes became very small, limiting the statistical power of the analyses. We think this is sufficiently addressed in the Discussion:

'...(in) the meta-analysis, evidence of protective effects was generally limited to higher levels of active commuting. The high 'doses' of utility cycling reported in previous studies are likely to be achieved when cycling journeys are taken frequently and consistently (e.g. twice daily, five days per week).'

8. The authors could do much more with the issue of cycling and the displacement of physical activity. It appears cycling doesn't seem to displace other forms of PA (this is important) and is mentioned for commuter cyclists. Could further analysis be done to argue this is the case, with the position being that encouraging cycle commuting actually achieves a net population gain in physical activity?

We agree that our finding that cycling, and utility cycling in particular, does not displace participation in other physical activity is important. It contributes to an increasing body of literature, including our own, which shows that utility physical activity is carried out in addition to recreational physical activity. This point is outlined in lines 285-290 of the Discussion. As we already show that commuter cycling was associated with participation in greater levels of total physical activity (Table 5), it is unclear what additional analyses of the available data would shed further light on the issue. In any case, given the

large number of analyses already presented in this paper we would prefer not to include more results.

9. A final point - I'm not sure it matters what type of cycling has the greatest health benefit and therefore should be promoted to the public. There are many environmental and transport/congestion reasons for encouraging all forms of cycling as I'm sure the authors are aware.

We thank the reviewer for this observation and agree that there are good reasons for encouraging all forms of cycling. This point is highlighted in both the Introduction ("Promoting cycling as an alternative to motorised transport would result in reduced carbon emissions, traffic congestion and noise pollution while providing people with an opportunity to integrate regular physical activity into their lives") and the Discussion ("In the meantime, our results suggest that even modest 'doses' of cycling may reduce mortality risk and do not suggest any evidence of an adverse effect, thereby contributing to the growing environmental, social and public health case for promoting cycling in individuals and populations").

10. Does intensity of effort make the difference on health or is it simply how long you do it for (mins per week)?

Please see our response to point 6 above.

11. Disappointing that levels of cycling were so low, and may be under the critical cut-point needed to demonstrate independent health impacts.

We agree that the levels of cycling were disappointingly low. East Anglia has the highest rates of cycling in the UK (Goodman, 2013), suggesting that rates may be even lower in other parts of the country. In terms of the amount of cycling needed to demonstrate independent health impacts, we have addressed this issue in response to a previous comment.

Additional References

Goodman A. (2013). Walking, cycling and driving to work in the English and Welsh 2011 Census: Trends, Socio-Economic Patterning and Relevance to Travel Behaviour in General. PLoS ONE 8 (8): e71790.

Wannamethee SG, Shaper AG, Walker M (1998). Changes in physical activity, mortality, and incidence of coronary heart disease in older men. *Lancet* 351: 1603-1608.

Paffenbarger RS Jr, Hyde RT, Wing AL, Lee I-M, Jung, DL, Kampert JB. (1993). The association of changes in physical activity level and other lifestyle characteristics with mortality among men. *N Engl J Med*, 328: 538-545.